# Network-driven anomalous transport is a fundamental component of brain microvascular dysfunction

Florian Goirand [1,2], Tanguy Le Borgne [1✉] & Sylvie Lorthois [2✉]

Blood microcirculation supplies neurons with oxygen and nutrients, and contributes to clearing their neurotoxic waste, through a dense capillary network connected to larger tree-like vessels. This complex microvascular architecture results in highly heterogeneous blood flow and travel time distributions, whose origin and consequences on brain pathophysiology are poorly understood. Here, we analyze highly-resolved intracortical blood flow and transport simulations to establish the physical laws governing the macroscopic transport properties in the brain micro-circulation. We show that network-driven anomalous transport leads to the emergence of critical regions, whether hypoxic or with high concentrations of amyloid-$\beta$, a waste product centrally involved in Alzheimer's Disease. We develop a Continuous-Time Random Walk theory capturing these dynamics and predicting that such critical regions appear much earlier than anticipated by current empirical models under mild hypoperfusion. These findings provide a framework for understanding and modelling the impact of microvascular dysfunction in brain diseases, including Alzheimer's Disease.

[1] University of Rennes, CNRS, Géosciences Rennes, UMR 6118 Rennes, France. [2] Institut de Mécanique des Fluides de Toulouse, UMR 5502, CNRS, University of Toulouse, Toulouse, France. ✉email: tanguy.le-borgne@univ-rennes1.fr; sylvie.lorthois@imft.fr

The brain microvascular network provides an efficient, highly integrated, and dynamic infrastructure for the distribution of blood[1–3]: it supplies oxygen, nutrients, and, if needed, drugs to all cells in the brain, and ensures the removal of their metabolic waste[4,5]. As the brain lacks any substantial energy reserve, it also acts as a short-term regulation system, which responds quickly and locally to the metabolic needs of neurons[2,6]. In ageing and disease, however, the progressive appearance of abnormal vessel architectures, including reduced capillary diameters or stalling, and the decrease in regulation efficiency together reduce blood flow and the availability of oxygen[2,5,7–11]. This also alters the clearance of metabolic waste, including neurotoxic forms of amyloid-$\beta$ centrally involved in the pathogenesis of Alzheimer's Disease (AD)[4,9,10]. Thus, understanding the links between the microvascular architecture, reduced blood flow, and impaired oxygen delivery and metabolic waste clearance is a key challenge to decipher the role of microvascular dysfunction in brain disease.

The microvascular architecture is structured by tree-like arterioles and venules that connect to a dense capillary network[3,12,13]. While this organization ensures a large surface of exchange between blood and the brain tissue, it also induces strong spatial heterogeneities of vessel flows and capillary transit times, leading to heterogeneous oxygenations[14,15]. Even in normal conditions, some vessels with low blood flow rates approach the hypoxic threshold[16,17]. These critical vessels may be particularly vulnerable to further pathological stress[17], consistent with the appearance of small hypoxic regions in the cortex of ageing and AD mice[9,18]. Since reduced capillary flow also compromises metabolic waste clearance, critical vessels with abnormally high intravascular concentrations of amyloid-$\beta$ may also be expected. Yet, it is unknown how such critical vessels may appear under normal conditions nor how they may progress in response to pathological stress, such as hypoperfusion[10,19,20].

In fact, the dynamics governing oxygen or amyloid-$\beta$ distributions in such networks are fundamentally non-local: the solute concentration in a given vessel depends on the blood travel time from penetrating arterioles, where blood enters into the brain cortex, to this vessel, which integrates all blood velocities along the corresponding pathways. Furthermore, the distribution of blood flow is also non-local, i.e., driven by the whole vascular architecture. Because the impact of such non-local dynamics on relevant network scale processes is difficult to resolve explicitly, the blood travel time distribution through the microvascular network has been represented by phenomenological models[21], including early indicator dilution analysis models[21–24] and the recent Capillary Transit Time Heterogenity (CTH) model[14]. These models rely on empirical blood travel time distributions, following mathematical functions chosen to match the experimentally observed distributions[25]. A key property emerging from CTH models is that increased transit time heterogeneities induce a decreased efficiency of oxygen supply[14]. Consistently, reduced transit time heterogeneity has been experimentally confirmed in the cortical layers with the highest levels of metabolic activity[26] or in response to neuronal activation[27] while increased transit time heterogeneity has been inferred, based on such models, from clinical imaging data in AD patients[28]. Besides these few examples, CTH models have been used to interpret a large amount of experimental data[15,18,29–31] thus helping to identify increased vascular heterogeneity as a key general mechanism of neuronal injury.

A fundamental limitation of current phenomenological models is that they do not quantitatively relate the transport dynamics to the underlying network architecture and flow distributions. This makes it difficult to understand and predict how changes in vessel architecture may influence blood travel time heterogeneity and thus alter oxygen supply and metabolic waste clearance. Furthermore, in vivo measurements are limited to time scales smaller than the blood recirculation time (~5 s)[32], which limits the range available for the calibration of empirical models. Hence, their predictive power for the statistics of longer time scales, which likely control the appearance of vessels with critical oxygen or amyloid-$\beta$ concentrations, is strongly dependent on the mathematical functions chosen to parametrize travel time distributions[22,23]. Yet, the physical mechanisms shaping these distributions and how they depend on the network structure[21] are poorly understood.

Theoretical analyses of transport in model random networks have shown that these systems can exhibit anomalous transport dynamics, i.e., characterized by slow power-law decays of the large travel time probabilities[33]. The latter has been successfully described by Continuous Time Random Walk (CTRW) theories, providing analytical expressions of travel time distributions as a function of the microscale structures and flow distributions[33–36]. Although microvascular networks fundamentally differ from such random networks, their complex structure potentially contains the fundamental ingredients for anomalous transport dynamics to develop. To explore this hypothesis, we use highly resolved simulations of blood flow in such networks, validated from in vivo measurements, that provide access to the full statistics of blood flow and transport dynamics in realistic microvascular networks. We use these insights to uncover the scaling laws of blood flow distributions arising from the microvascular architecture and develop an effective transport model at the scale of the network that captures these properties. This provides analytical solutions for the blood travel time distributions inferred from the physics of transport in these networks. Our model predicts that the interplay between the spatial distribution of arterioles and venules and the mesh-like architecture of the capillary bed[12] leads to the emergence of anomalous transport dynamics. This implies that the occurrence probability of large blood travel times is significantly larger than predicted by current models. We couple this model to the kinetics of oxygen consumption and amyloid-$\beta$ production in brain cells to show that these anomalous transport properties control the early development of critical vessels with low oxygen or large amyloid-$\beta$ concentrations under hypoperfusion. These findings hence provide a framework to measure, understand and model the onset and development of brain diseases, such as AD[37].

## Results
Blood travel times through microvascular networks, i.e., from any network inlet to any network outlet, are also referred to as traversal times[24] or, simply, as transit times[14], whereas the distribution of travel times is sometimes referred to as the impulse response function, as the transfer function of the system or as the microvascular response function[38]. To avoid any ambiguity, we use below the terms travel times for transport across the whole network and transit times for transport across a single vessel (as defined by Eq. (2)).

**Vessel flow rates and transit times follow broad distributions**. Our analysis is based on highly resolved simulations of blood flow in anatomic microvascular networks, validated by comparison with in vivo measurements (see Methods and Supplementary Note 1). We first present the results obtained in a microvessel network (~15,000 vessels) digitized from a 1 mm³ of the mouse cortex (see Supplementary Figure S1) and then compare the results to another mouse intracortical microvessel network and to bio-mimetic networks (Supplementary Notes 3 and 8). The dense capillary bed in the first network is homogeneous and space-

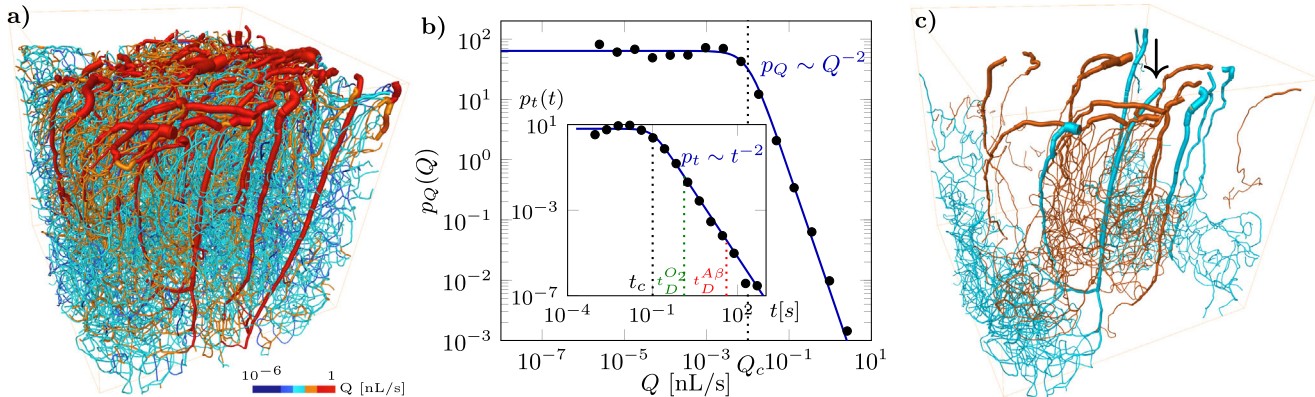

**Fig. 1 The blood flow organization in cortical microvessel networks induces broadly distributed flow rates and vessel transit times. a** Simulated blood flow map in a microvascular network feeding $1\,mm^3$ of the mouse cortex (see Supplementary Figure S1a, b and Supplementary Movie S1). Vessel flow rates are represented with blue shades for $Q < Q_c$ and red shades for $Q > Q_c$ (see also Supplementary Figure S1c, d, f and Supplementary Movie S2). **b** PDF of simulated flow rates (black dots) compared with the approximation of Eq. (1) (continuous blue line). The characteristic flow rate $Q_c$ separating the uniform and power-law regimes are indicated as a dotted line. Inset: PDF of vessel transit times. The diffusion coefficients for oxygen and amyloid-$\beta$ ($D_{O_2} = 2 \times 10^{-9}\,m^2\,s^{-1}$ and $D_{A\beta} = 6 \times 10^{-11}\,m^2\,s^{-1}$) yield diffusion times ($t_D^{O_2}$ and $t_D^{A\beta}$) indicated by the green and red dashed lines, respectively. **c** Example of trajectories visiting <30 vessels (orange) and >70 vessels (blue), originating from the arteriole shown by the arrow (see Supplementary Figure S2a–d and Supplementary Movies S3).

filling[17,39], with a narrow distribution of vessel diameters ($4.8 \pm 0.9\,\mu m$). The network is fed and drained by ~15 arteriolar and ~30 venular trees (Fig. 1a, Supplementary Figure S1 and Supplementary Movie S1). Simulations integrate the non-linear blood rheology and red blood cell repartition at diverging vessel bifurcations (see Methods). The average flow rate is $\langle Q \rangle_0 = 4 \times 10^{-2}\,nL/s$. The probability density function (PDF) of blood flow rates in vessels shows a large spread across about seven decades around the average flow rate (Fig. 1b). This PDF exhibits two different regimes: it is uniform in the low flow rate range and decays as a power law with exponent $-2$ above a characteristic value $Q_c = 10^{-2}\,nL/s$. The number of vessels in these regimes is approximately $P(Q < Q_c) = 60\%$ and $P(Q > Q_c) = 40\%$, with large flow rates developing preferentially in the neighborhood of arterioles and venules (Fig. 1a and Supplementary Figure S1f, Supplementary Movie S2).

The flow rate PDF is well approximated by a Cauchy distribution (Fig. 1b), consistent with the theoretical asymptotic behavior obtained in large Random Regular Graphs[40].

$$p_Q(Q) = \frac{2}{\pi Q_c}\frac{1}{1 + (Q/Q_c)^2}, \quad (1)$$

A direct consequence of the broad distribution of flow rates $Q$ is the broad distribution of advective transit times in vessels (inset of Fig. 1b), which are defined as:

$$t = \frac{l\pi d^2}{4Q}, \quad (2)$$

where $l$ is the vessel arc length and $d$ is the vessel diameter. As the flow rate PDF, the vessel transit time PDF is characterized by two regimes separated by the characteristic time $t_c = \frac{\langle l \rangle_{cap}\pi\langle d \rangle_{cap}^2}{4Q_c} = 10^{-1}$ s, where $\langle l \rangle_{cap} = 50\,\mu m$ and $\langle d \rangle_{cap} = 5\,\mu m$ are taken as characteristic capillary length and diameter, respectively. Since the vessel length $l$ and diameter $d$ vary weakly compared to the flow rate $Q$, the transit time variability is mainly driven by the flow rate fluctuations. We thus estimate the PDF of vessel transit times from the PDF of flow rates as $p_t(t) = p_Q(Q)dQ/dt$, also yielding a Cauchy distribution

$$p_t(t) = \frac{2}{\pi t_c}\frac{1}{1 + (t/t_c)^2}, \quad (3)$$

in good agreement with the simulations (inset of Fig. 1b). The scaling $p_t(t) \sim t^{-2}$ for long times, $t > t_c$, is induced by the uniform flow rate PDF $p_Q(Q) \sim cst$ at low flow rates, $Q < Q_c$, leading to $p_t(t) \sim dQ/dt \sim t^{-2}$ (Eq. (2)). Equation (3) implies that the probability of a vessel belonging to this power-law regime is $p(t > t_c) = \int_{t_c}^{\infty} dt p_t(t) = 1 - 2/\pi\tan^{-1}(1) = 0.5$. Hence, half of the vessels belong to this regime. Since our anatomical networks have about $N = 15{,}000$ vessels, the lowest probability that we can measure is $1/N = 7e^{-5}$. This corresponds to a maximum transit time $t_m = t_c\tan(\pi/2(1 - 1/N)) \approx 10^4 t_c$. Therefore, the size of the network allows observing this power-law behavior across up to four orders of magnitudes in transit times from $t_c$ to $10^4\,t_c$ (see inset of Fig. 1b). This power-law scaling yields a non-negligible probability of extremely long vessel transit times (e.g., 1% of vessels have a transit time $t > 50\,t_c = 5\,s$), which may lead to the emergence of anomalous transport properties at the network scale[33,34].

This vessel transit time PDF $p_t(t)$ characterizes the vessel transport statistics in the absence of diffusion. Diffusion introduces a maximum cutoff time $t_D = \langle l \rangle^2/D$ in the transit time PDF, corresponding to the diffusive transport time over a vessel length. For oxygen and amyloid-$\beta$, the two species considered here, the range of times $t_c < t < t_D$ over which the power-law $p_t(t) \sim t^{-2}$ holds thus covers respectively one and three orders of magnitude (Fig. 1b, inset). Therefore, this power-law regime affects significantly more amyloid-$\beta$ clearance than oxygen supply.

**Network trajectory lengths and travel times show anomalous transport statistics.** In addition to the broad distribution of vessel transit times, solute transport at the scale of the microvascular network is also controlled by the distribution of trajectory lengths from arterioles to venules. Our particle tracking simulations (see Methods, Fig. 1c and 2a and Supplementary Movie S4) show that trajectories lengths $L$, expressed in the number of visited vessels, vary from <10 to ~80 (Fig. 2b). The trajectory length PDF is characterized by a power-law scaling $p_L(L) \sim L^{-2}$ between two characteristic lengths $L_0 = 12$ and $L_c = 50$. Above $L_c$, the PDF decays sharply as $p_L(L) \sim \exp(-L/L^*)$ with $L^* = 5$ (Fig. 2b):

$$\begin{cases} p_L(L) \sim L^{-2} & L_0 < L \le L_c \\ p_L(L) \sim \exp(-L/L^*) & L > L_c \end{cases} \quad (4)$$

The scale $L^*$ characterizes the exponential decay of the PDF of trajectory lengths at large lengths and is estimated by fitting Eq.

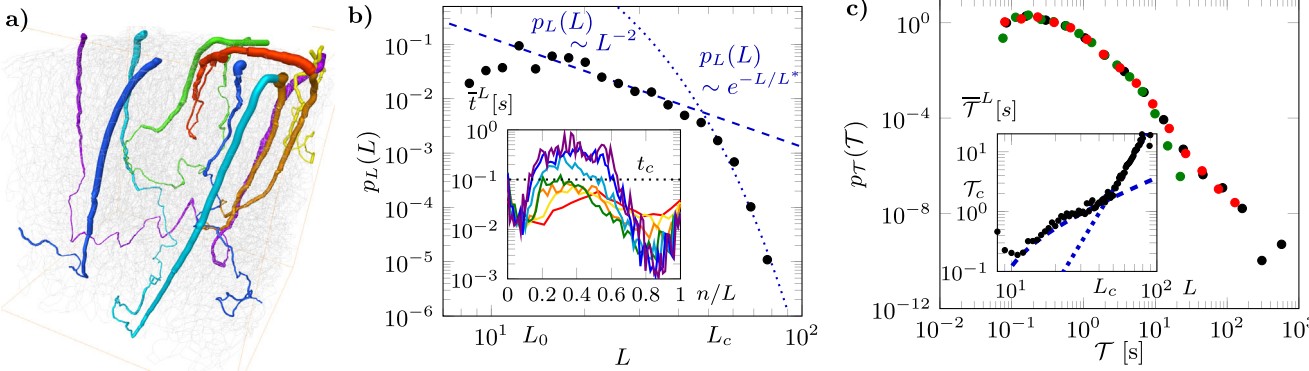

**Fig. 2 Trajectory lengths and travel times are broadly distributed at network scale. a** 3D view of typical particle trajectories (see also Supplementary Movie S4) with growing numbers of visited vessels (Red: $L = 20$; Yellow: $L = 30$; Orange: $L = 40$; Green: $L = 50$; Cyan: $L = 60$; Blue: $L = 70$; Violet: $L = 80$). **b** Probability Density Function (PDF) of trajectory lengths. Inset: Local vessel transit times $\overline{t}^l$ averaged over all trajectories of same length $L$ as a function of the number $n$ of vessels visited since the inlet arteriole, normalized by $L$. Results for different trajectory lengths are shown with the same color conventions as in **a**. **c** PDF of travel times through the network for purely advective (black dots), oxygen (green dots), and amyloid-$\beta$ (red dots) transport. Inset: Average travel time $\overline{\mathcal{T}}^L$ as a function of trajectory length $L$ (see also Supplementary Note 4 and Supplementary Figure S4a, b). The linear and power-law tendencies (Eq. (5)) are shown respectively as dashed and dotted blue lines. Note that the linear tendency does not appear as a straight line in logarithmic coordinates because of the constant $\mathcal{T}_0$ in Eq. (5) (See Supplementary Note 4).

(4) to the simulated PDF (Fig. 2b). This cutoff in trajectory lengths is likely controlled by a complex interplay between the topology of the network and the density of venules/arterioles, which prevents arbitrary large trajectories to develop.

Averaging over multiple trajectories of equal length, we computed the evolution of the local average transit time along the trajectory, as a function of the number $n$ of vessels visited since the inlet arteriole (Fig. 2b, inset). These average local transit times $\overline{t}^L$ are of the order of $10^{-2}$ s close to the inlet arterioles, then increase, up to two orders of magnitudes for the longest trajectories, then decrease again in the vicinity of the outlet venules. The network travel time averaged over all trajectories of equal length $\overline{\mathcal{T}}^L$ thus increases with the trajectory length $L$ following two different trends (Inset of Fig. 2c, Supplementary Note 4 and Supplementary Figure S4a, b),

$$
\begin{cases}
\overline{\mathcal{T}}^L(L) \approx \tau_1(L - L_0) + \mathcal{T}_0 & L_0 < L \le L_c \\
\overline{\mathcal{T}}^L(L) \approx \mathcal{T}_c \left(\frac{L}{L_c}\right)^4 & L > L_c
\end{cases}
\tag{5}
$$

where $\mathcal{T}_0 = 0.2$ s, $\mathcal{T}_c = 1.8$ s, and $\tau_1 = 0.04$ s is on the order of the mean transit time over particle trajectories.

The broad distribution of vessel transit times, together with the distribution of trajectory lengths, leads to a broad range of travel times at the network scale $p_\mathcal{T}(\mathcal{T})$ (Fig. 2c). Without diffusion, simulated travel times vary over four orders of magnitudes. Accounting for the diffusive cutoff transit time (inset of Fig. 1b), the travel time distribution still covers over two orders of magnitude for oxygen and three orders of magnitude for amyloid-$\beta$ (Fig. 2c).

**Flow and transport properties emerge from the physics of dipole flows in networks**. The statistical properties driving these transport dynamics can be understood as arising from different topological properties of the flow field, as schematized in Fig. 3. The measured scaling $p_Q(Q) \sim Q^{-2}$ is a characteristic of dipole flows[41] (see Supplementary Note 3, Supplementary Figure S3d). In the present system, high flow rates are localized around arterioles and venules that act as multiple sources and sinks (Fig. 1a and Supplementary Figure S1, Supplementary Movie S1), driving the flow in the network. Hence, in the large flow range, $Q > Q_c$, the flow field behaves statistically as a superposition of

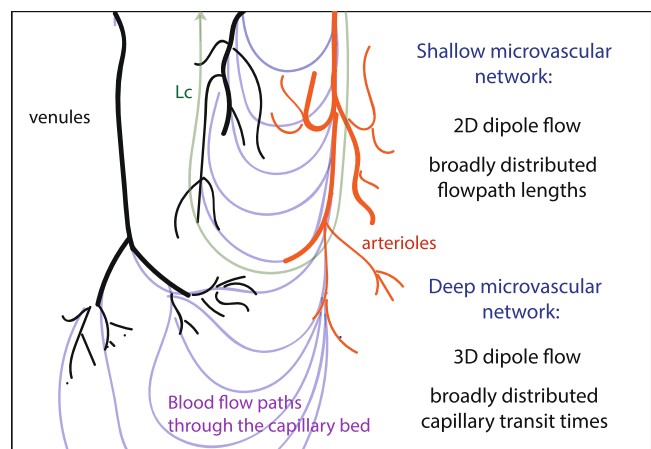

**Fig. 3 Schematic illustration of the blood flow organization in the cortical microcirculation inferred from the observed flow and transport properties.** Arterioles and venules are represented in red and black respectively. Blood flow paths through the capillary bed are represented as light purple lines. The characteristic trajectory length $L_c$ is the maximum length of trajectories that can travel directly from one arteriole to a neighboring venule (green line).

dipoles. At low flow rates, $Q < Q_c$, the flow rate statistics differ from the continuous dipole model and become uniform, a signature of the network structure (see Supplementary Note 3, Supplementary Figure S3d). In random networks, the flow rate PDF is indeed theoretically expected to follow exponential distributions, driven by the random additions and divisions of flow at vertices[42]. In the low flow rate range, this asymptotically leads to a uniform distribution of flows. Here, the low flow vessels are far from arterioles and venules. Therefore, their statistics are dominated by the random fluctuations induced by the network structure. Noteworthy, similar flow rate PDFs are observed for dipole flows on space-filling networks with architectures of increasing complexity, see Supplementary Figure S3b–d. Thus, the observed flow rate statistics are generic and arise from the interplay between a structured dipole-like topology for large flow rates and random network topology for small flow rates.

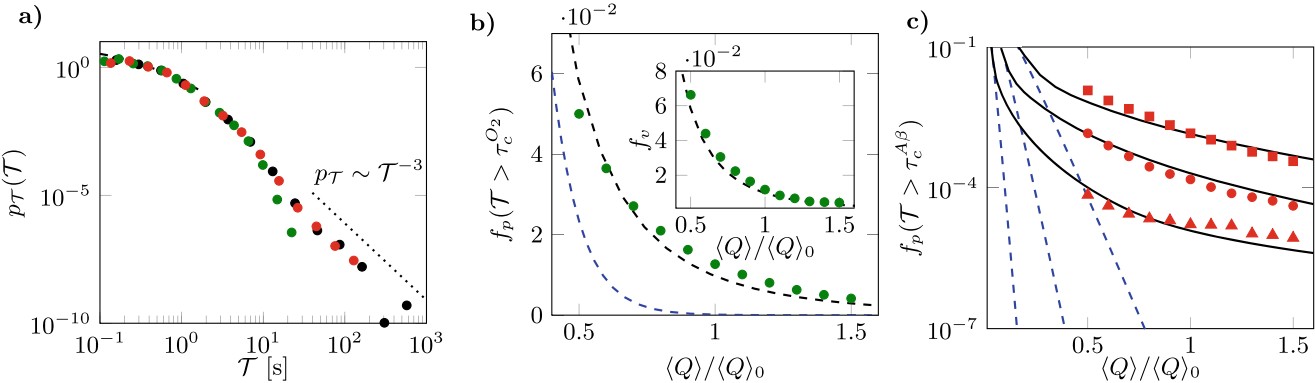

**Fig. 4 Long travel times, captured by our mechanistic model, drive the emergence of critical areas with impaired oxygen delivery and amyloid-β clearance under hypoperfusion. a** CTRW model predictions (full black line) compared to purely advective, amyloid-β, and oxygen transport simulations (black, red, and green dots, respectively). The prediction of the mean-field transport model (Eq. (6)) and of the reference CTH model (Supplementary Note 7) are shown as dashed black and blue lines, respectively. **b** Fraction $f_p$ of network travel times larger than $\tau_c^{O_2}$ for oxygen delivery as a function of average flow rate $\langle Q \rangle / \langle Q \rangle_0$. Simulations (green dots) are compared to the predictions of the mean-field model (dashed black line) and of the reference CTH model (dashed blue line). The inset shows the fraction $f_v$ of vessels in the network that are only reached by flow paths with travel times to these vessels larger than $\tau_c^{O_2}$, as a function of the flow rate. **c** Fraction $f_p$ of network travel times larger than $\tau_c^{A\beta} = 8$ s (squares), $\tau_c^{A\beta} = 16$ s (dots) and $\tau_c^{A\beta} = 40$ s (triangles), as a function of average flow rate for amyloid-β clearance. The predictions of the CTRW model (Eq. (7)) for each value of $\tau_c^{A\beta}$ are shown as continuous black lines. The predictions of the reference CTH model (see Supplementary Note 7) for each value of $\tau_c^{A\beta}$ are shown as blue dashed lines.

The distribution of trajectory lengths (Fig. 2b), is also consistent with this dipole flow interpretation (see Supplementary Note 3, Supplementary Figure S3e). For dipole flows on a finite-size network, the power-law regime $p_L(L) \sim L^{-2}$ develops until a cutoff length, corresponding to the network size, which sets the maximum trajectory length. In the present system, the characteristic trajectory length $L_c$ that sets the transition to the exponential cutoff (Eq. (4)) is the maximum length of trajectories that can travel directly from one arteriole to a neighboring venule[43] (Fig. 1c and Supplementary Figure S2a, b, Supplementary Movie S3 (orange trajectories)). These trajectories typically reach the bottom of the simulated domain and include on average 20 arteriolar steps, 20 venular steps, and 10 capillary steps in between (see Fig. 3 and Supplementary Note 2, Supplementary Figure S2f). Longer trajectories connect more distant arterioles and venules via the deep capillary bed, where most vessels are in the low flow regime ($Q < Q_c$) (Figs. 1c, 3 and Supplementary Figure S2c, d, Supplementary Movie S3 (blue trajectories)).

The relationship between average travel times and trajectory lengths $\overline{\mathcal{T}}^L(L)$ (Eq. (5)) shows a transition from a linear to a power-law scaling (Inset of Fig. 2c and Supplementary Figure S4a, b). In the first linear regime, the number of visited capillaries remains approximately constant when the trajectory length increases (see Supplementary Note 2 and Supplementary Figure S2f). This implies that the additional visited vessels belong to arterioles and venules. The linear scaling of $\overline{\mathcal{T}}^L(L)$ indicates that the average transit time $t$ in such vessels remains approximately constant when trajectories explore deeper sections of the network. As arterioles penetrate at depth, their flow rate decreases but so does their diameter and length, which may explain this finding. This may constitute an evolutionary advantage contributing to spatially uniformizing the supply and clearance of solutes across the network while minimizing total dissipation and blood volume[44–46]. In the second regime, the power-law scaling $\overline{\mathcal{T}}^L(L) \sim L^4$ is characteristic of dipole flow in 3D systems (see Supplementary Note 3): in the deep network, blood flows from the pre-capillary arterioles to the post-capillary venules through a complex 3D network of capillaries (Fig. 1c, 3, Supplementary Figure S2c, d and Supplementary Movie S3 (blue trajectories)) where the number of visited capillaries increases steeply with the trajectory length (Supplementary Figure S2f).

**Mean-field transport dynamics are governed by dipolar trajectory length distributions.** The distribution of trajectory lengths (Eq. (4)), coupled with the relationship between average travel time and trajectory length (Eq. (5)), provides a mean-field transport model for the travel time PDF across the network (Supplementary Note 4). This mean-field description, which neglects random fluctuations due to the network structure but captures the dipole-driven trajectory length distribution, is characterized by a transition from power law to a stretched exponential behavior:

$$p_{\overline{\mathcal{T}}}(\overline{\mathcal{T}}) \sim \left( (\overline{\mathcal{T}} - \mathcal{T}_0)/\tau_1 + L_0 \right)^{-2}$$
$$\text{for } \overline{\mathcal{T}} < \mathcal{T}_c$$
$$p_{\overline{\mathcal{T}}}(\overline{\mathcal{T}}) \sim \left( \frac{\overline{\mathcal{T}}}{\mathcal{T}_c} \right)^{-3/4} \exp\left( -\frac{L_c}{L^*} \left( \frac{\overline{\mathcal{T}}}{\mathcal{T}_c} \right)^{1/4} \right) \quad (6)$$
$$\text{for } \overline{\mathcal{T}} > \mathcal{T}_c$$

with $\mathcal{T}_c = 1.8$ s (Eq. (5)). This model captures the travel time distribution over the first two orders of magnitude (Fig. 4a and Supplementary Note 4, Supplementary Figure S4c). We compare this prediction to that of a reference CTH model, that assumes a Gamma distribution of travel times, whose parameters are calibrated from in vivo data[14] (see Supplementary Note 7). Since experimentally measured travel time distributions are limited to times smaller than ~5 s due to blood recirculation[27,32], this CTH model serves here as a reference to assess the effect of neglecting the experimentally inaccessible longest travel times. While it captures relatively well the shape of the travel time distribution in the low range, the reference CTH model significantly underestimates the probability of late times (Fig. 4a). Accounting for the trajectory length distribution via the mean-field model allows capturing a significant part of this long-time dynamics. The mean-field model, however, does not capture the power-law behavior of the longest travel times, driven by vessels with transit times $t > t_c$ in the deep capillary network (Fig. 1c). Since the oxygen cutoff diffusion time $t_D^{O_2}$ is close to $t_c$ (inset of Fig. 1b), this long-time regime does not affect much oxygen transport, which is well represented by the mean-field model (Fig. 4a).

**Random flow fluctuations in the capillary network control large blood travel times.** To obtain a full description of the transport dynamics, in particular for low diffusivity solutes such

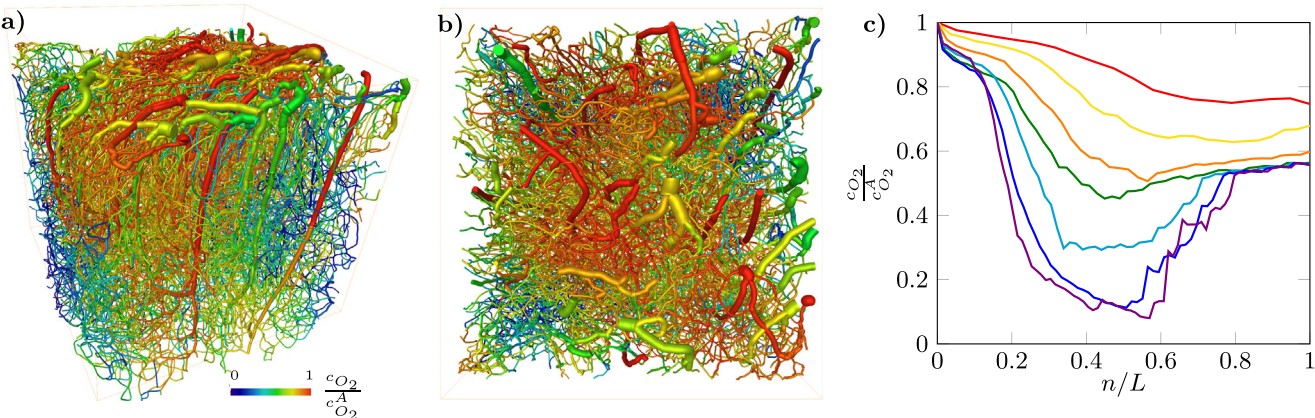

**Fig. 5 Simulated distribution of blood oxygen concentration relative to oxygen arteriolar concentration in the microvascular network represented in Fig. 1. See also Supplementary Movie S5. a** Side view. **b** Top view. **c** Evolution of the mean vessel oxygen concentration along trajectories of size $L$, concentrations are normalized by the arteriolar concentration $c_{O_2}^A$. Red: $L = 20$; Yellow: $L = 30$; Orange: $L = 40$; Green: $L = 50$; Cyan: $L = 60$; Blue: $L = 70$; Violet: $L = 80$.

as amyloid-$\beta$, we seek an effective transport model that captures the heterogeneity of vessel transit times around the average behavior described above. Fluid elements transported along a given trajectory in the network move from one vessel to the next with a broadly varying time step (see Supplementary Notes 5 and 6 and Supplementary Figure S5a)). This closely corresponds to the conceptual framework of CTRW[34,36], which has been proved relevant as an effective representation of transport in random networks[33,35]. However, as a consequence of the dipole structure of the flow at a large scale, two characteristics of the transport dynamics differ from a conventional CTRW representation. First, the number of steps in a given trajectory is broadly distributed according to trajectory lengths (Eq. (4)). Second, the mean local transit time at the nth step within a trajectory of length $L$ depends on the trajectory length (Eq. (5), inset of Fig. 2b and Supplementary Note 4). Fluctuations of local transit times around these mean local transit times define a noise term that is independent of trajectory lengths and follows a power-law scaling controlled by the flow distribution (see Supplementary Note 5). We, therefore, developed a CTRW framework capturing these two properties (see Supplementary Note 6).

This framework allows deriving an analytical solution for the travel time distribution $p_{\mathcal{T}}(\mathcal{T})$ in the network,

$$p_{\mathcal{T}}(\mathcal{T}) = p_{\mathcal{T}}^1(\mathcal{T}) + p_{\mathcal{T}}^2(\mathcal{T}), \qquad (7)$$

where $p_{\mathcal{T}}^1(\mathcal{T})$ and $p_{\mathcal{T}}^2(\mathcal{T})$ are defined in Laplace space as,

$$\tilde{p}_{\mathcal{T}}^1(s) = e^{-s\mathcal{T}_0} \sum_{L=L_0+1}^{L_c} p_L(L) \big( \mathcal{P}(s\tau_1) \big)^{L-L_0} \qquad (8)$$

and

$$\tilde{p}_{\mathcal{T}}^2(s) = e^{-s\mathcal{T}_0} \mathcal{P}^{L_c-L_0}(s\tau_1) \sum_{L=L_c+1}^{\infty} p_L(L) \mathcal{P}^{L-L_c}\left( s \frac{\mathcal{T}_c(L/L_c)^4 - \mathcal{T}_c}{L-L_c} \right) \qquad (9)$$

where $s$ is the Laplace variable, $\mathcal{P}(s) = 2sK_2(2\sqrt{s})$ with $K_2$ the Bessel function of the second kind and $p_L(L)$ is given by Eq. (4).

This CTRW model is fully determined from the trajectory length PDF (Eq. (4)) and the relationship between average time and trajectory length (Eq. (5)). It provides an accurate prediction over a broad range of travel times with no fitting parameter (Fig. 4a). In particular, it captures the late-time power-law decay:

$$p_{\mathcal{T}}(\mathcal{T}) \sim \mathcal{T}^{-3}. \qquad (10)$$

The emergence of this power law is consistent with the CTRW

theory that predicts stable power-law distributions for noises characterized by power-law exponents equal or smaller than 3 (see Supplementary Figure S5)[34,36]. Hence, the late-time transport behavior is characterized here by stable anomalous transport, driven by the broad distribution of flow rates in the network. As far as we know, no experimental data have been obtained to confirm this result in brain microcirculation. However, injections of flow-limited tracers in the coronary network of isolated rabbit hearts, which avoids blood recirculation, yielded a power-law decay of the late-time regime with exponent $-3.21 \pm 0.27$[47]. This suggests that, despite the variability of microvascular architecture between organs[48,49], the combination of tree-like structures with a dense capillary network in their global organization is sufficient to drive this late-time power-law decay.

**Simulation of three-dimensional oxygen distribution in microvascular networks.** Assuming that oxygen consumption in vessels follows first-order kinetics[25] (see Methods and Supplementary Note 9), we use our particle tracking simulations to compute three-dimensional fields of oxygen concentration in vessels (Fig. 5a, b and Supplementary Movie S5). The characteristic reaction time of the first-order kinetics $\tau_r^{O_2} = 1.5$ s is estimated by matching the simulated ratio between oxygen concentrations at venular outlets and those at arteriolar inlets to typical measured values (see Methods). The evolution of the oxygen concentration along trajectories, averaged over trajectories of equal length (Fig. 5c), is in qualitative agreement with previous experimental observations[17]. Oxygen decays close to the network inlets, as trajectories penetrate down in the cortex, then reaches a minimum in the capillary bed (except for the shortest trajectories $L = 20$), and increases again as blood flows up to the venular outlets (Fig. 5c). The minimum oxygen value along trajectories decreases with the trajectory length $L$, as longer trajectories penetrate deeper in the network. This directly results from the increase of blood travel time with increasing trajectory length (Eq. (5) and inset of Fig. 2c). The minimal oxygen concentration in the network, reached in the deep capillaries of the longest trajectories, is close to 1/12 of the inlet concentration, i.e., 10 mm Hg of oxygen partial pressure, assuming 120 mm Hg at the inlets. This value is typically used to identify hypoxic brain regions in animal experiments[50,51]. Hence, our results are consistent with the recent experimental observation that some vessels approach the hypoxic threshold in the cortex of normal mice[17]. The increase of oxygen towards the outlet venules is also consistent with in vivo observations showing an increase of oxygen

concentrations with increasing venous diameters[17]. Although its origins are debated[17], the two minimal ingredients included in our simulations, i.e., first-order decay of oxygen with travel time and mixing at vessel intersections, are sufficient to capture this behavior. Hence, the travel time and trajectory length statistics explored here are key to explaining the oxygen dynamics.

**Anomalous transport drives the early appearance of hypoxic regions under conditions of hypoperfusion.** Hypoperfusion, i.e., the decrease of the average blood flow, is a major pathological stress associated with many diseases[19], including early stages of AD[20,37]. Schematically, it may be due to reduced perfusion pressure or increased cerebrovascular resistance, e.g., induced by capillary occlusions[8] or reduced vessel diameters[9] in AD. In the latter case, the total resistance of the capillary bed has been estimated to increase more than threefold, leading to a ~50% blood flow reduction. This large hypoperfusion level significantly increases tissue hypoxia[9] and is believed to favor amyloid-$\beta$ accumulation in the brain[10], thus participating in disease progression. However, whether lower levels of hypoperfusion (5–30%), such as induced by capillary occlusions[8], may be involved in disease onset is debated[10]. In this context, we use our modeling framework to quantify the appearance of hypoxic regions, focusing on the occurrence probability of trajectories with travel times above the critical travel time $f_p(\mathcal{T} > \tau_c^{O_2})$, which corresponds to the hypoxic threshold $c_{O_2} = 1/12 c_{O_2}^A$ [50,51]. Using the first-order kinetic model (Supplementary Note 9, equation S28), we estimate this critical travel time to be $\tau_c^{O_2} = \log(12)\tau_r^{O_2} = 3.7$ s. The probability of exceeding $\tau_c^{O_2}$ along a given flow pathway is derived from our transport model as $f_p(\mathcal{T} > \tau_c) = \int_{\tau_c}^{\infty} d\mathcal{T} p_{\mathcal{T}}^{\langle Q \rangle}(\mathcal{T})$, where $p_{\mathcal{T}}^{\langle Q \rangle}(\mathcal{T})$ is the predicted network travel time distribution for any flow rate $\langle Q \rangle$. The latter is derived from the travel time distribution in normal conditions as $p_{\mathcal{T}}^{\langle Q \rangle}(\mathcal{T}) = p_{\mathcal{T}}^{\langle Q \rangle_0}(\mathcal{T}\langle Q \rangle_0/\langle Q \rangle)$, where, $\langle Q \rangle_0$ is the blood flow in normal conditions, deduced from the above simulations, for which all parameters, including the perfusion pressure, correspond to physiological data (see Methods and Supplementary Note 1). In other words, changing the average flow rate $\langle Q \rangle$ is equivalent to rescaling the critical time as $\tau_c \langle Q \rangle / \langle Q \rangle_0$. This also holds true when the flow rate change is induced by capillary occlusions as the shape of the travel time PDFs is similar, not only for different sets of mouse microvascular networks, but also when up to 10% of capillaries are occluded (see Supplementary Note 8 and Supplementary Figure S6).

Under normal perfusion, the fraction of travel times larger than $\tau_c^{O_2}$ is equal to $f_p(\mathcal{T} > \tau_c^{O_2}) = 1.3\%$ (Fig. 4b). This fraction increases non-linearly as the average flow decreases, e.g., by a factor 4 when $\langle Q \rangle$ decreases by a factor 2, an evolution accurately captured by the mean-field model (Eq. (6)). As discussed above, oxygen transport is mostly controlled by the dipole-driven transport regime and its travel time distribution is relatively well represented by the mean-field transport model (Fig. 4a). We thus approximate $p_{\mathcal{T}}^{\langle Q \rangle_0}$ by the mean-field travel time distribution (Eq. (6)), which provides an accurate prediction of the simulated fraction of critical travel times for all flow rates (Fig. 4b). Since the latter integrates the broad distribution of trajectory lengths, it predicts a significantly larger occurrence probability $f_p(\mathcal{T} > \tau_c^{O_2})$ than the reference CTH model (Fig. 4b), which is two orders of magnitude below the simulated probabilities under baseline perfusion. Noteworthy, the fractions of trajectories $f_p(\mathcal{T} > \mathcal{T}')$ measured at the outlet are very close to the fractions of critical vessels $f_v(\mathcal{T} > \mathcal{T}')$ within the network, i.e.,

vessels that are only visited by fluid elements taking a time larger than $\mathcal{T}'$ to reach them from the inlet arterioles (Supplementary Note 2, Supplementary Figure S2e). Hence, the travel time statistics at the outlet venules offer a surrogate for the transport statistics within the network. As a result, the mean-field model also provides a good prediction for the probability of occurrence of hypoxic vessels within the network (inset of Fig. 4b). When hypoperfusion is induced by capillary occlusions, these hypoxic vessels may likely add up to the occluded vessels, thus enhancing their impact at the early stages of AD[8].

**The weak diffusivity of amyloid-$\beta$ amplifies the impact of anomalous transport.** We use a similar modeling method as for oxygen to relate the blood travel time statistics to amyloid-$\beta$ concentrations (see Methods and Supplementary Note 10). Owing to its low diffusivity, amyloid-$\beta$ is highly sensitive to random flow fluctuations in the capillary network as discussed above (Fig. 2a). Therefore we approximate $p_{\mathcal{T}}^{\langle Q \rangle_0}$ by the CTRW model (Eqs. (7–9)). The metabolism and neurotoxicity of amyloid-$\beta$ involve multiple soluble and insoluble isoforms and are still poorly understood[52–54]. To account for this uncertainty, we consider a range of critical times $\tau_c^{A\beta} = 8$, 16, or 40 s. This leads respectively to a three-, five- or tenfold arterio-venous increase of the total intravascular amyloid concentration (see Methods and Supplementary Note 10), which is much larger than the measured physiological increase of ~20% [55], thus yielding different degrees of compromised clearance. As expected, the probabilities of occurrence of trajectories with travel times above these critical times $f_p(\mathcal{T} > \tau_c^{A\beta})$ vary significantly with $\tau_c$ (Fig. 4c). Although the CTH model predicts an exponential evolution of the fraction of critical travel times $f_p(\mathcal{T} > \tau_c^{A\beta})$ with $\langle Q \rangle$ (linear in semi-log in Fig. 4c), the CTRW model captures the clearly non-exponential trend observed from the simulations. This is a signature of the power-law scalings of travel time distributions, driven by anomalous transport. Hence, the reference CTH model underestimates the fractions of travel times with inefficient amyloid-$\beta$ clearance by several orders of magnitude, while the CTRW model accurately predicts the probability of these critical travel times for all $\langle Q \rangle$ and $\tau_c$.

## Discussion

By reducing the complexity of the transport problem in anatomically realistic networks while keeping the essential physics of transport emerging from the network architecture, our analysis reveals the physical mechanisms by which the microvascular architecture shapes the blood travel time distribution. This is a major fundamental open question in microvascular physiology and a bottleneck for accurate quantification of hemodynamic parameters from brain imaging data in a broad range of applications, from clinical studies aimed at improving the diagnosis and/or staging of brain disease to fundamental studies on cerebral blood flow and metabolism, neurovascular coupling, cerebral autoregulation and/or blood–brain barrier function in health and disease[21].

We have demonstrated that the blood travel time distributions are driven by two fundamental mechanisms constitutive of anomalous transport dynamics[34]: broadly distributed blood trajectory length and broadly distributed capillary transit times. As schematized in Fig. 3, the former is determined by dipolar flow patterns resulting from the localized connections with upstream and downstream surface vessels, while the latter is driven by random-like fluctuations within the capillary network. For high diffusivity species, such as oxygen, travel time distributions are cutoff by diffusive transport, a third fundamental mechanism that

dampens the random-like fluctuations. These fundamental insights yield the first physics-based analytical solutions for transport at the scale of cortical microvascular networks, accurately predicting the statistical distributions of travel times of different solutes, including oxygen (Eq. (6) and amyloid-$\beta$ (Eqs. (7–9))).

This offers an alternative to current effective models of transport at the scale of microvascular networks. Such models either consider simplified networks as combinations of parallel elements (e.g., [56]), overlooking the link between the transport dynamics and the underlying microvascular architecture, or use empirical functions (e.g., [14]). The latter are calibrated from in vivo measurements, which, due to blood recirculation, are limited to travel times of 5s. Extrapolation to larger time scales is constrained by the underlying parametrization chosen for the distribution of travel times, most often gamma distributions[14,25,57]. The resulting exponential decay underestimates the probability of large travel times that follow a slow power-law decay induced by the anomalous transport dynamics uncovered here. Such models predict that a significant level of hypoperfusion, where blood flow is reduced by ~20% compared with normal perfusion, should be reached for hypoxic vessels to appear. Even in normal conditions, we found that a small proportion (~1.2%) of hypoxic vessels develop in the microvascular network. This finding is consistent with experimental measurements[16,17]. Furthermore, accounting for anomalous transport leads to a regular non-linear increase of the proportion of hypoxic vessels with the decreased flow. The impact of anomalous transport is stronger for amyloid-$\beta$, which has a smaller diffusion coefficient. Hence, the probabilities of occurrence of critical vessels with inefficient metabolic waste clearance is orders of magnitude larger than predicted by current empirical models under normal conditions and their increase under hypoperfusion occurs much earlier than anticipated by these models.

Overall, these findings underpin the physical mechanisms by which moderate levels of hypoperfusion yield a non-linear expansion of hypoxic regions and lead to increased accumulation of amyloid-$\beta$ in the brain tissue. Crucially, hypoxia and amyloid-$\beta$ accumulation are two key ingredients of the amyloid cascade, the positive feedback loop linking hypoperfusion and amyloid-related pathways of AD (see, e.g., Fig. 3 in ref. [10]). Combined with the recent discovery that, at early stages of AD, every single capillary occlusion has a similar, and cumulative, impact on blood flow, without any threshold effect[8], this suggests that capillary occlusions, even in small proportions, may trigger this positive feedback loop, leading to pericyte activation and capillary constrictions. Current numerical simulations[8] neglecting these pathological stresses as well as any compensatory effects, e.g., variations of the cerebral perfusion pressure, tend to underestimate the magnitude of hypoperfusion compared to experiments in animal models of AD. Hence, we speculate that pathological stresses induced by the amyloid cascade have an outsized impact on disease progression. Our results suggest that such cascading events may be initiated by spatial heterogeneities of blood flow. In particular, the critical regions with abnormally high amyloid-$\beta$ concentration, which can be interpreted as hotspots for amyloid species accumulation, are deterministically located in regions fed by long trajectories. This could possibly explain why no association between capillaries with abnormal/stalled flow and amyloid deposition has been found experimentally[8]. Furthermore, as larger travel times are expected for trajectories feeding the sub-cortical regions, this may explain their specific vulnerability[58,59], as highlighted by the appearance of white matter hyper-intensities in clinical imaging, in both cerebrovascular disease and AD[28,60]. This suggests that, while the capillary bed is a continuously adapting network, which remodels to reduce the extent of hypoxic regions[61], strong constraints may be imposed by the architecture of the vascular network and the associated locations of inlets and outlets. Besides shedding light on the impact of capillary occlusions at early stages of AD, and more generally on the role of hypoperfusion in AD onset and progression, these findings contribute to explaining the considerable overlap between vascular and neurodegenerative factors in the pathogenesis of brain disease[2,7].

These results will also contribute to improving the quantification of physiological parameters from brain perfusion or functional imaging data, whether acquired by optical imaging, computed tomography, or magnetic resonance. Such quantification generally relies on the choice of mathematical functions to represent the distribution of intravascular travel times below the scale of imaging resolution. Our findings hence establish the physical grounds for defining these travel time functions and relate them to the microvascular architecture. This may help to account for vascular alterations, which are currently overlooked when interpreting human clinical imaging data in patient populations[62], contributing to bridging the gap with knowledge acquired from animal experiments.

The analysis of transport dynamics in highly-resolved simulations of blood flow in anatomically realistic microvascular networks is complementary to in vivo experiments in that it provides access to the full flow and transport statistics, opening a window to deciphering the underlying non-local physics. With the fast progress of numerical simulations and imaging capacities, the presented framework may be further improved. For example, larger simulations domains, possibly up to the whole mice brain (see, e.g., [63]) may be considered. The diphasic nature of blood or more realistic reactive intravascular transport dynamics (see, e.g., [64]), simplified here as first-order kinetics and neglecting the oxygen-binding cooperativity to hemoglobin, could also be considered in the future, as well as transport and metabolic processes within the brain tissue (see, e.g., [25,54,65]).

Our theoretical framework hence opens perspectives for the development of predictive, physics-based, transport models at the scale of brain microvascular networks that account for the complexity of microvascular architectures. The resulting scaling laws are generic to a large variety of networks, from simplified ones to accurate anatomical representations, with or without capillary occlusions. This suggests that the uncovered anomalous transport mechanisms are general, even if the parameter values and pre-factors of scaling laws may be slightly dependent on the specific assumptions in our blood flow computational scheme. Thus, the variability of the network architecture, including differences between brain areas and between species[39,66], differences due to long term vessel remodeling in hypoxia, aging or disease[61,62,67], as well as passive or active diameter variations resulting from changes in pressure, blood flow, brain autoregulation and/or neurovascular coupling[2,6,21,68], is unlikely to fundamentally alter the nature of the statistical laws that we have derived. Blood is ultimately transported in the brain capillary network, the structure of which is highly similar between species[39], so that velocity fluctuations are expected to follow similar distributions as described here. Interestingly, the predicted late-time power-law decay of travel time probabilities, with exponent −3, has even been observed with a similar exponent in the coronary microcirculation[47], despite the much stronger variability of microvascular architecture between organs[48,49]. Hence, anomalous transport induced by velocity fluctuations in the capillary network provides an alternative fundamental mechanism for this scaling, previously interpreted as arising from an underlying fractal organization of blood flow[21,69].

## Methods

**Brain microvascular networks.** We use a first large postmortem data set (~15,000 vessel segments in a ~1 mm³ region) from the mouse vibrissa primary sensory (vS1) cortex obtained by[3,70] (see Supplementary Note 1 and Supplementary Movie S1) and previously used for simulation studies by[8,43,64,71,72]. Details on the procedures used for correcting the vessel's diameters to match the in vivo distributions and for classifying vessels into arterioles, capillaries, and venules are given in the Supplementary Material from[8]. For comparison, we use a second dataset from the same cortical region, obtained from another mouse[3,70], as well as dense, space-filling networks of increasing complexity, including randomly generated networks, which reproduce well the spatial and functional statistics of the mouse capillary bed[39] (Supplementary Notes 3 and 8).

**Simulation of blood flow and Lagrangian transport.** Blood flow is modeled using a non-linear network approach accounting for the complex rheology of blood in the microcirculation[8,73], with prescribed boundary conditions (see Supplementary Note 1 for details on methods and validation procedures). This yields the pressure $P$ at each vertex and the flow rate $Q$ and haematocrit $H$ in each vessel. We then simulate the transport of $5 \times 10^7$ passive Lagrangian particles, injected from arteriolar inlets with a probability proportional to their flow rate. We advect these particles at local average vessel velocity ($v = \langle Q \rangle / \langle Q \rangle_0 Q/\pi r^2$), where $\langle Q \rangle / \langle Q \rangle_0$ represents the global variation of cerebral blood flow compared with physiological baseline conditions. We distribute them in downstream vessels using flux-weighted fractions, i.e., with a probability proportional to the local flow rate (Supplementary Note 2). For each particle trajectory, we computed the corresponding trajectory length $L$, i.e., the total number of visited vessels, the Lagrangian transit time series, i.e., the succession of local advective transit times $t$ from vessel to vessel in the trajectory (Fig. S5a, Supplementary Note 5), as well as the travel time $\tau$, i.e., the sum of these local transit times. The effect of diffusion on intravascular transport is taken into account by introducing a maximum transit time equal to the diffusion time over the vessel length: the local transit times $t$ is set equal to the local diffusion times ($t_D = l^2/D$) if $t_D < t_a$, where $t_a$ is the local vessel advection time. Note that to compute the advection time, we considered the average velocity in each vessel and did not resolve the Poiseuille flow profile in the direction transverse to the flow in each vessel. Resolving these velocity profiles within capillaries does not change much the travel time distributions and only slightly modifies the early travel times[64]. Hence, at the network scale, the intravessel flow variability is negligible compared to the inter-vessel variability.

All numerical procedures for blood flow simulations have been implemented in a custom-built C++ code[74] and the post-processing algorithms for the Lagrangian analysis have been developed in Python (V2.7).

**First-order kinetics models of oxygen supply and amyloid-β clearance.** In order to evaluate the characteristic times associated with oxygen supply and amyloid-β clearance, and following[14,54,75,76], we assume that they can both be described by the following first-order kinetics models (see Supplementary Notes 9 and 10):

$$\frac{\partial c_{O_2}}{\partial t} = -k_{O_2} c_{O_2} \tag{11}$$

and

$$\frac{d(c_{A_\beta})}{dt} = \frac{d(c_{A_\beta} - c_{A_\beta}^T)}{dt} = -k_{A_\beta}(c_{A_\beta} - c_{A_\beta}^T) \tag{12}$$

where $k_{O_2}^{-1} = \tau_r^{O_2}$ and $k_{A_\beta}^{-1} = \tau_{A_\beta}$ are, respectively, the characteristic times for oxygen consumption in the brain tissue and for amyloid-β clearance.

Coupling these kinetics with the travel time distribution $p(\mathcal{T})$, we obtain the ratio between oxygen concentration at the outlet (venules), $c_{O_2}^V$, and at the inlet (arterioles), $c_{O_2}^A$,

$$c_{O_2}^V / c_{O_2}^A = \int_0^\infty d\mathcal{T} \exp\left(-k_{O_2}\mathcal{T}\right) p_{\mathcal{T}}(\mathcal{T}) \tag{13}$$

In the same way, we obtain the ratio between the amyloid-β concentration at the outlet (venules), $c_{A_\beta}^V$, and its tissue concentration, $c_{A_\beta}^T$,

$$c_{A_\beta}^V / c_{A_\beta}^T = 1 - \int_0^\infty d\mathcal{T} \left\{ \left(1 - \frac{c_{A_\beta}^A}{c_{A_\beta}^T}\right) \exp\left(-k_{A_\beta}\mathcal{T}\right) p_{\mathcal{T}}(\mathcal{T}) \right\} \tag{14}$$

Using the measured travel time distribution for oxygen (Green dots in Fig. 2c), we match Eq. (13) to the typical resting oxygen extraction fraction of ≈30%[14], which gives $c_{O_2}^V / c_{O_2}^A \approx 0.7$[14], yielding $\tau_r^{O_2} = k_{O_2}^{-1} = 1.5$ s. Interestingly, this time is of the same order as the decay time measured in a single cell in vitro measurements of RBC oxygen desaturation dynamics[77], equal to 800 ms. Similarly, using typical amyloid-β concentration values (see, e.g.,[54,78]) and an arterio-venous increase of 20%[55] yields $\tau_r^{A_\beta} = k_{A_\beta}^{-1} = 97$ s. This estimated time is of the same order as can be deduced by abluminal-to-luminal permeability measurements in an in vitro blood–brain barrier model (hCMEC/D3 endothelial monolayers)[79], which yielded

a permeability $P \simeq 18 \times 10^{-5}$ cm/min. Assuming an endothelial thickness of ~1 µm, this leads to a characteristic time of 33 s. By contrast, previous estimates based on a compartmental model yielded $\tau_r^{A_\beta} = 3000$ s[76], i.e., two orders of magnitudes above the in vitro results. Hence, despite the simplified reaction kinetics considered here, capturing the full range of travel times appears to be a key element for modeling reactive processes in the cortex.

**Reporting summary.** Further information on research design is available in the Nature Research Reporting Summary linked to this article.

## Data availability
The flow simulation data analyzed in this manuscript and the corresponding Lagrangian trajectories have been made available on Harvard Dataverse (https://doi.org/10.7910/DVN/QJDUUA).

## Code availability
The C++ flow simulation code and the custom Python (V2.7) post-processing algorithms are available within 2 months from the corresponding authors upon request subject to a nonexclusive, revocable, non-transferable, and limited right to use for research and evaluation purposes only, excluding any commercial use.

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

## Acknowledgements

Research reported in this publication was supported by the European Research Council under ERC grant agreements 615102 (BrainMicroFlow) and 648377 (ReactiveFronts) and by the NIH (awards R21CA214299 and 1RF1NS110054). We gratefully acknowledge P. Blinder, P. Tsai, and D. Kleinfeld for sharing anatomical networks, C.B. Schaffer and N. Nishimura for inspiring discussions, N. Nishimura for critically reading a previous version of this manuscript, and M. Berg who developed the network flow solver. The funders had no role in the study design, data collection, and analysis, decision to publish, or preparation of the manuscript.

## Author contributions

All authors contributed equally to the conceptual aspects of this work and to manuscript writing. F.G. developed the numerical methods and analyzed the data. T.L.B. and S.L. designed the research, jointly supervised the work, and contributed to data analysis and interpretation.

## Competing interests

The authors declare no competing interests.
