## [Peer Review File · Nature Communications]

Network-driven anomalous transport is a fundamental component of brain microvascular dysfunctionREVIEWER COMMENTS

Reviewer #1 (Remarks to the Author):

About once per year, I get sent a paper for review that I wish that I had written myself. This is one such paper. A number of years ago, I thought about how to solve this problem and wasn't able to make much progress, purely due to the complexity of the networks and trying to formulate a bridging approach from considering every vessel to one that considered a whole network. The authors have formulated a rigorous and elegant solution to the problem of transport times in complex vascular networks (even if they call them travel times!) I found this a fascinating paper and think it a very valuable contribution to the literature as we seek to get a better quantitative understanding of the processes that govern blood flow and oxygen transport in the brain.

I do have some comments on the manuscript, which are below.

1. Probably my main comment is that the results are very heavily based on a single network and a single simulation of flow through this network (with a specific set of imposed boundary conditions). This is perfectly reasonable, but I think it should be acknowledged more explicitly. Other networks and simulations (or even just different boundary conditions) might give different results. [I did also notice that the title of Figure 1 refers to 'networks', when there is in fact only one!]
2. I was very interested in the tail of the distribution (the 'non-negligible probability' mentions below Equation 3). My slight quibble with this is that the network has a finite number of vessels, so the extreme values are (1) exceptionally unlikely to occur; and (2) unlikely to have been modelled by this finite network. Essentially the tail of the distribution has no evidence for its existence without a very much larger set of simulations being used. I would just like to see something mentioned here as this being a limitation to the details of the distribution.
3. Some of the material surrounding the transport of oxygen is a little speculative, but I found it pretty persuasive and very thought-provoking. Quite how the model can be validated remains an open question, but this study is really valuable in helping to point the way to important (future) experimental studies. The only thing that I would add would be that the capillary bed is a continuously-adapting network, so it seems likely that the network will adapt to reduce any regions of hypoxia (there has been a lot of work on capillary network re-organisation by Secomb). I would incline just to mention this somewhere - and to highlight that the model is a purely passive one, i.e. it doesn't consider the active regulation of flow.

However, these are all relatively minor comments and I commend the authors on this excellent study. The manuscript is very clearly written and will be an important addition to the literature.

Reviewer #2 (Remarks to the Author):

The manuscript „Network-driven anomalous transport is a fundamental component of brain microvascular dysfunction“ by Goirand et al. describes a new framework for modelling blood travel times in cerebro-vascular systems and the use of it to explore the potential cause of some brain diseases. The manuscript is well-written and easy to understand. The model is based on a 1 microliter cube of a mouse cortex and consists of impressing 15,000 vessel segments. The importance of vessel architecture has been shown before, but this work shows some aspects and impacts on travel time heterogeneity in a realistic network structure for the first time.

There are some limitations of your approach, which are not well represented and discussed in the manuscript.

- The use of a static scan of vessel network: you assume that length and diameter of the capillaries do

not change significantly with flow rate. Do you have a reference for this assumption? While this assumption might be realistic, I am wondering how dynamic effects like vasodilation on arterioles and vessel recruitment of capillaries (e.g. Brendan C Fry, Tuhin K Roy, Timothy W Secomb: Capillary recruitment in a theoretical model for blood flow regulation in heterogeneous microvessel networks, *Physiol Rep* . 2013 Aug;1(3):e00050. doi: 10.1002/phy2.50) would affect your considerations. The conclusion "We have demonstrated that the blood travel time distributions are driven by two fundamental mechanisms constitutive of anomalous transport dynamics" is only valid for this static network case.

- Simulation of oxygen and amyloid-beta based on diffusion coefficient. Did you consider tortuosity of the vessel in your calculations? While oxygen shows a diffusive behaviour, amyloid-beta will definitely not for an intact blood-brain barrier (there might be carrier processes involving LRP1 which supports exchange, though). Diffusive exchange of amyloid-beta will only be possible for a compromised BBB. Indicating that your model allows deeper understanding of amyloid-beta clearance is an over-statement.
- I understand that you calculate a single velocity for each vessel. Did you simulate a flow profile in the vessels at any time? Do you have an estimate how dispersion will affect your findings?
- How does the non-fluid behaviour of haematocrit in capillaries influence the results?

Sorry for my ignorance, but I find the term local transit time not clearly defined. Is this the transit time of a single vessel? This largely varies with vessel length and the prognostic/diagnostic value of such a definition is not completely clear to me. I understand the relevance of travel times where it would be most efficient if the overall travel times would be the same for all possible pathways. How does this link to the relevance of local transit time?

Minor:

Eq.4: Do you have an explanation for $L^*=5$? How did you find this number? Which other parameters does it depend on?

While all other videos played fine, supplementary movie S4 did not.

Reviewer #3 (Remarks to the Author):

This microvascular physiology-modelling study by the authors Goiran, Le Borgne and Lorthois, from a biological perspective, is an thorough and excellent piece of work on the origins of the microvascular flow heterogeneity in mouse capillary network. In contrast to the classical simplified microvascular simulations used to evaluate the capillary flow patterns and their impact on tissue oxygenation, this model is based on a real and 3-D capillary network and incorporates many physical features of complicated biphasic blood flow dynamics. This study has a great potential to be expanded in the near future for understanding microvascular dysfunction in many acute and chronic neurological diseases and can also be applicable to other tissues outside the nervous system. These results give insight about the origins and impact of the dynamic capillary flow irregularities, like temporary stalls observed in normal animals, that have been shown to be exacerbated in animal models of stroke and Alzheimer's disease. The extent of the capillary transit variation is highly interesting and also provides clues about the selective vulnerability of deep cortical tissues to microvascular insults. I believe, in overall, this paper is a great contribution to the field.

I just have a few simple comments to improve the discussion of findings.

How applicable are these results to the human microvessel network? A brief discussion on the similarities and differences between mouse and human capillary network architecture can be added to the text to improve the translational value of this work.

When modeling the oxygen concentration distribution, was different levels of metabolic activity (hence oxygen consumption) across different cortical layers considered? If not, in what way could this modulate the results?

A strong emphasis is made on Alzheimer pathogenesis, as it is understandable that increased capillary stalling was observed in a mouse AD model and amyloid-beta toxicity is implicated in AD. However, in the paper by Cruz-Hernandez et al, Nat Neuro 2019, PMID 30742116) there was no association between capillaries with abnormal/stalled flow and amyloid deposition. Here, the capillary transit and amyloid beta clearance simulations seem to indicate capillary flow-dynamic hot spots for amyloid species accumulation. What are the authors' comments on this difference? Moreover, microvascular architecture seems to be changing over the course of AD (as reviewed in Steinman et al, PMID: 33536876). I believe these should be added to the discussion and cited, also discussed how these morphological changes could affect the predictions of capillary flow trajectories and oxygen distribution.

Should be added to the methods: Which programming platform was used for modeling and mathematical calculations? And are the models open to the scientific community upon request?

REPLY TO REVIEWER 1

- [1] *“About once per year, I get sent a paper for review that I wish that I had written myself. This is one such paper. A number of years ago, I thought about how to solve this problem and wasn’t able to make much progress, purely due to the complexity of the networks and trying to formulate a bridging approach from considering every vessel to one that considered a whole network. The authors have formulated a rigorous and elegant solution to the problem of transport times in complex vascular networks (even if they call them travel times!) I found this a fascinating paper and think it a very valuable contribution to the literature as we seek to get a better quantitative understanding of the processes that govern blood flow and oxygen transport in the brain.”*

We sincerely thank the Reviewer for his/her very positive assessment of our work.

- [2] *“I do have some comments on the manuscript, which are below.*

Probably my main comment is that the results are very heavily based on a single network and a single simulation of flow through this network (with a specific set of imposed boundary conditions). This is perfectly reasonable, but I think it should be acknowledged more explicitly. Other networks and simulations (or even just different boundary conditions) might give different results. [I did also notice that the title of Figure 1 refers to ‘networks’, when there is in fact only one!]”

Our manuscript may not have been clear enough regarding this question but, in fact, we used two different anatomical data-sets extracted from normal mouse brains, as detailed in the Methods Sections (with the specific set of boundary conditions described in Supplementary Information SIA). We showed in Section H of the Supplementary Information that the results are independent of the specific network used, and provided explanations about why the size of the considered networks is sufficient for that purpose. Furthermore, we also used several model networks of increasing complexity (illustrated in Supplementary Information Fig. SI3a and b). These included random bio-mimetic networks, i.e., which reproduce the morphological, topological and functional properties of the intracortical capillary bed of normal mice (Smith et al. *Frontiers in Physiology*, 2019, Ref. 39 in our manuscript). For these different model networks, we imposed single or multiple dipolar injections and

used no-flow boundary conditions on the bottom and lateral faces, instead of the pseudo-periodic boundary conditions used in the mouse networks. In all cases, we retrieved the same long time travel-time statistics, i.e. well described by a power law with exponent -3. This general result is consistent with our theoretical derivations showing these late statistics are dominated by the network-induced fluctuations captured by our mechanistic model (CTRW model).

Figure R1: Flow rate distribution in model networks with simple boundary conditions.

a) Snapshot of the flow distribution in a bio-mimetic micro-vascular network with logarithmic color scale. As boundary conditions we impose different uniform pressures at the left and right faces and no flow on other faces. **b)** Probability Density Function of flow rates (dashed line) and exponential distribution (full line). In agreement with the theory of Alim et al. 2017 (reference 42 in the manuscript), the flow rate PDF is close to an exponential distribution, which leads to the plateau at low flow rates, also observed in our mouse anatomical networks. Figure adapted from Goirand, "Statistical modeling of blood flow and transport in brain microvascular networks", PhD Thesis, Université de Toulouse (defended June 18 2021).

In particular, we have tested the effect of boundary conditions by simulating flow in random networks with simple boundary instead of the multiple dipolar injections used in anatomical networks. We thus imposed different uniform pressures on the left and right faces of the domain and no flow conditions on other faces (Fig. R1a). As shown in Fig. R1b, this lead to the same plateau in the flow rate PDF at low flow rates, the key property leading to the late-time power law behavior of the travel time distribution. Hence, the transport properties revealed in our study are generic to a large range of networks, and independent

of the specific boundary conditions imposed. They are inherent to the network nature of microvascular systems, as predicted from our theory. Therefore, although a larger amount of high-quality anatomical dataset will likely be available in the future, the basic properties that we have uncovered will persist.

Change made: We have clarified the fact that several mouse networks have been used in the Results (see p. 3) and Methods (see p. 13) Section of the main text and in Section A of the Supplementary Information. We have also clarified the boundary conditions imposed in model networks in Section C of the Supplementary Information.

- [3] *“I was very interested in the tail of the distribution (the ‘non-negligible probability’ mentions below Equation 3). My slight quibble with this is that the network has a finite number of vessels, so the extreme values are (1) exceptionally unlikely to occur; and (2) unlikely to have been modelled by this finite network. Essentially the tail of the distribution has no evidence for its existence without a very much larger set of simulations being used. I would just like to see something mentioned here as this being a limitation to the details of the distribution.”*

This question is of course legitimate. However, the size of our networks is largely sufficient to robustly confirm the existence of the late-time power law statistics of capillary transit times. The Probability Density Function (PDF) of capillary transit times follows $p_t(t) = \frac{2}{\pi t_c} \frac{1}{1+(t/t_c)^2}$, with $t_c = 0.1$ s (see Eq.(3) and inset of Fig.1b). This implies that the probability of a vessel to belong to the power law regime is $p(t > t_c) = \int_{t_c}^{\infty} dt p_t(t) = 1 - 2/\pi \tan^{-1}(1) = 0.5$. Hence, half of the vessels belong to this regime. Since our anatomical networks have about $N = 15,000$ vessels, the lowest probability that we can measure is $1/N = 7e^{-5}$. This corresponds to a maximum transit time $t_{max} = t_c \tan(\pi/2(1 - 1/N)) \approx 10^4 t_c$. Therefore, the size of the network allows observing this power law behavior over about four orders of magnitudes in time, from t_c to $10^4 t_c$, as shown in our manuscript (see inset of Fig.1b).

Change made: We have detailed these arguments below equation 3 in the revised manuscript.

- [4] *“Some of the material surrounding the transport of oxygen is a little speculative, but I found it pretty persuasive and very thought-provoking. Quite how the model can be validated remains an open question, but this study is really valuable in helping to point the way to important (future) experimental studies. The only thing that I would add would be that the capillary bed is a continuously-adapting network, so it seems likely that the network will adapt to reduce*

any regions of hypoxia (there has been a lot of work on capillary network re-organisation by Secomb). I would incline just to mention this somewhere - and to highlight that the model is a purely passive one, i.e. it doesn't consider the active regulation of flow."

Figure R2: **Comparison of flow and transport statistics when considering heterogeneous or uniform conductances.** a) Flow rate PDF. b) Vessel transit time PDF. c) Network travel time PDF. a-c) Scaling laws are computed for a micro-vascular network with uniform conductances (circles) and for an anatomical one (dots). Quantities with tilde sat on top are rescaled by their average value. The blue dashed line in panel c shows the \mathcal{T}^{-3} for long network travel times.

Indeed, brain microvascular networks are continuously adapting, with short-term diameter variations of about 20 to 30% induced by neurovascular coupling and long-term changes induced by vessel remodelling. However, transit time fluctuations induced by such diameter variations are expected to be negligible compared to transit time heterogeneity generated by the random network structure, which spans about 6 orders of magnitude (inset of Fig. 1b). Indeed, the capillary transit time PDF (equation 3), derived from the flow rate PDF (equation 2) by neglecting diameter variations, captures very well the simulated transit times (see inset of Fig.1b). To strengthen this argument, we have computed the flow and transit time statistics in an anatomical network with imposed uniform vessel diameter (see Fig. R2 below). The flow, transit time and travel time statistics were found to be very similar to those obtained in our original network. Hence this confirms that diameter fluctuations do not affect significantly transport properties, which are primarily governed by the network structure.

Change made: We have revised the end of the discussion (see p. 13 of the revised manuscript) to better outline that the vascular network may actively adapt its diameters and remodel,

and have commented that these fluctuations are expected to have minor effects in the travel time distributions. We have added a relevant reference to Secomb and collaborators on hypoxia-driven remodelling (Secomb et al. PLOS Comp Biol 2013, ref. 61). We have also added the new figure R2 in Section H of the Supplementary Information together with the arguments developed above.

REPLY TO REVIEWER 2

- [1] *“The manuscript ”Network-driven anomalous transport is a fundamental component of brain microvascular dysfunction” by Goirand et al. describes a new framework for modelling blood travel times in cerebro-vascular systems and the use of it to explore the potential cause of some brain diseases. The manuscript is well-written and easy to understand. The model is based on a 1 microliter cube of a mouse cortex and consists of impressing 15,000 vessel segments. The importance of vessel architecture has been shown before, but this work shows some aspects and impacts on travel time heterogeneity in a realistic network structure for the first time.”*

We thank the Reviewer for his careful reading of our work and his relevant comments and questions, which we answer below.

- [2] *“There are some limitations of your approach, which are not well represented and discussed in the manuscript.*

The use of a static scan of vessel network: you assume that length and diameter of the capillaries do not change significantly with flow rate. Do you have a reference for this assumption? While this assumption might be realistic, I am wondering how dynamic effects like vasodilation on arterioles and vessel recruitment of capillaries (e.g. Brendan C Fry, Tuhin K Roy, Timothy W Secomb: Capillary recruitment in a theoretical model for blood flow regulation in heterogeneous microvessel networks, Physiol Rep . 2013 Aug;1(3):e00050. doi: 10.1002/phy2.50) would affect your considerations. The conclusion “We have demonstrated that the blood travel time distributions are driven by two fundamental mechanisms constitutive of anomalous transport dynamics“ is only valid for this static network case. ”

Yes, we do simulate flow and transport in static scans of vessel networks. Including dynamic effects induced by active regulation mechanisms or vessel remodelling in such networks would

render the simulation and theoretical framework considerably more complex. Furthermore, as discussed below, temporal fluctuations in vessel diameters, which are on the order of 20 to 30%, have a negligible impact on transport dynamics compared to the network driven flow heterogeneity, which spans over 6 orders of magnitude (Fig. 1b). Thus, while including these dynamic effects would be one of the promising extension of our work, we do not consider them in this study in order to focus on the first order physics of transport emerging from the network architecture.

An important point in this discussion is first to clarify what we meant when writing “*Since the vessel length l and diameter d vary weakly compared to the flow rate Q , the transit time variability is mainly driven by the flow rate fluctuations*” in the paragraph before Eq. 3. This does not mean that vessel length and diameter could not be actively tuned depending on the flow rate, but only that their variability is much smaller than the variability of the flow rate within the network. The later spans across more than about six decades, as displayed in Fig. 1b. By contrast, diameters and length span less than two decades: the shortest length is $\sim 10\mu m$ and the largest one $\sim 500\mu m$ while the smallest diameter is $\sim 3\mu m$ and the largest one $\sim 50\mu m$. This variability is even considerably reduced when only considering capillary vessels. As a result, it is possible to straightforwardly deduce the distribution of vessel transit times from the distribution of flow rates, using Eq. 3. This has been verified in our manuscript by comparing the prediction of Eq.3 to the distribution of transit times deduced from the numerical results (continuous line versus dots in the inset of Fig 1b). Any modification of the vessel length or diameters by active regulation mechanisms or vessel remodelling would not change the above orders of magnitude, and would therefore not significantly change the conclusions of our work.

To give quantitative support to the above argument, we have performed additional numerical simulations in which the network structure (connectivity matrix) and boundary conditions are unchanged, but all conductances have been modified to have an equal value. This results in major changes of the distribution of conductances, or, as a results, of vessel diameters, which are much larger than any conceivable physiological or pathological passive or active vessel diameter variation. Yet, the computed flow rate, transit time and travel time distributions were almost unchanged compared to the anatomical simulations (see Fig. R3 below). Therefore, while active regulation mechanisms or vessel remodelling may slightly modify locally the flow rates and transit times, these changes are not expected to affect the

network-driven transport dynamics that we have uncovered in our study.

Figure R3: **Comparison of flow and transport statistics when considering heterogeneous or uniform conductances.** **a)** Flow rate PDF. **b)** Vessel transit time PDF. **c)** Network travel time PDF. **a-c)** Scaling laws are computed for a micro-vascular network with uniform conductances (circles) and for an anatomical one (dots). Quantities with tilde sat on top are rescaled by their average value. The blue dashed line in panel c shows the \mathcal{T}^{-3} for long network travel times.

Change made: We have revised the Discussion Section to better highlight that dynamic effects are unlikely to significantly change our conclusions (see end of the discussion on p. 13 of the revised manuscript). We have also added the new figure R2 in Section H of the Supplementary information together with the arguments developed above.

- [3] *“Simulation of oxygen and amyloid-beta based on diffusion coefficient. Did you consider tortuosity of the vessel in your calculations? While oxygen shows a diffusive behaviour, amyloid-beta will definitely not for an intact blood-brain barrier (there might be carrier processes involving LRP1 which supports exchange, though). Diffusive exchange of amyloid-beta will only be possible for a compromised BBB. Indicating that your model allows deeper understanding of amyloid-beta clearance is an over-statement.”*

There are two different elements in the question: accounting for vessel tortuosity is relevant for intra-vascular transport while modelling solute exchange through the blood-brain barrier (BBB) concerns extra-vascular transport. As explained in the manuscript (see Methods and Sections I and J in the Supplementary Information), we use diffusion coefficients to model intra-vascular transport but not to model solute transfer through the blood-brain barrier. The latter are modeled using first-order kinetics, both for oxygen or amyloid.

For **intra-vascular transport**, we did consider tortuosity to calculate the effect of diffusion on intra-vascular transport in our simulations. The diffusive vessel transit time for a given solute is computed as $t_D^i = l_i^2/D$, where D is the intra-vascular molecular diffusion coefficient of this specific solute and l_i is the arc length of vessels (and not the Euclidian distance between end vertices). The computed intra-vascular travel times accounting for both advection and diffusion along a given blood flow trajectory visiting L vessels is defined as $\mathcal{T} = \sum_{i=1}^L \min(t_a^i, t_D^i)$, with t_a^i is the vessel advective transit time. It thus correctly accounts for the impact of vessel tortuosity on intra-vascular transport within the network.

Concerning the **extra-vascular transport mechanisms**, we agree with the reviewer that, while oxygen shows a diffusive behaviour through the BBB, amyloid- β does not. Therefore we did not model amyloid- β transfer through the BBB using a diffusion model. As recently reviewed by Hladky and Barrand (Fluids and Barriers of the CNS, 2018), the transport mechanisms of amyloid- β are complex and still poorly understood. They involve LRP1 dependent transport, reaction of amyloid- β with soluble factors (apoJ, apoE, sLRP1) as well as at least four endocytotic/transcytotic systems. However, experimental studies using radio-labeled or other tracers to study the clearance of amyloid- β from brain to blood have consistently shown an exponential decrease of concentration with time at the scale of the brain, not only in mice models of AD (e.g. Shibata et al. J Clin Invest. 2000, ref. 76 in our MS, Swaminathan, JCBFM 2018, ref. 79 in our MS) but also in humans (e.g. Bateman, Nature Medicine 2006, Mawuenyega, Science 2010), suggesting that first-order kinetics is a good approximation for this process. Similar results have been obtained at the local scale with an in vitro blood-brain barrier model based on hCMEC/D3 endothelial monolayers (Ref Swaminathan, JCBFM 2018, ref. 79 in our MS). First-order kinetic descriptions of amyloid- β transport are thus commonly used in pharmacokinetic models (see e.g. Potter et al., Science Translational Medicine 2013, ref. 54 in our manuscript). This body of literature hence strongly supports our assumption of a first-order kinetics for amyloid- β transport through the blood-brain barrier.

References :

Hladky, Stephen B., and Margery A. Barrand. «Elimination of Substances from the Brain Parenchyma: Efflux via Perivascular Pathways and via the Blood–Brain Barrier». Fluids and Barriers of the CNS 15, n° 1.

Bateman, Randall J, Ling Y Munsell, John C Morris, Robert Swarm, Kevin E Yarasheski,

and David M Holtzman. «Human Amyloid- β Synthesis and Clearance Rates as Measured in Cerebrospinal Fluid in Vivo». *Nature Medicine* 12, n° 7.

Mawuenyega, K. G., W. Sigurdson, V. Ovod, L. Munsell, T. Kasten, J. C. Morris, K. E. Yarasheski, and R. J. Bateman. «Decreased Clearance of CNS -Amyloid in Alzheimer’s Disease». *Science* 330, n° 6012.

Change made: Concerning vessel tortuosity: After Eq. 2 and Eq. S1, we have clarified that l_i is the arc length of vessel i , so that our blood flow computations and calculations of transit times t_i , diffusive transit times t_D^i and travel times account for the tortuous nature of vessels.

With regard to amyloid transport through the BBB, we have emphasized more clearly that we use a first order kinetics models for amyloid- β transport and referred to the literature cited above. For that purpose, we have included the above details at the beginning of Section J of the Supplementary information, and cited the corresponding references.

- [4] “*I understand that you calculate a single velocity for each vessel. Did you simulate a flow profile in the vessels at any time? Do you have an estimate how dispersion will affect your findings?* ”

Yes, we considered the average velocity in each vessel and did not resolve the poiseuille flow profile in the direction transverse to the flow in each vessel. Indeed, one of us recently showed (Berg et al. 2020, ref. 64 in the manuscript) that resolving these velocity profiles within capillaries did not change much the travel time distributions and only modified slightly the early travel times. Hence, intra-vessel flow variability is negligible compared to the inter-vessel variability.

Change made: We have included the argument discussed above and the reference to Berg et al. 2020 in the revised manuscript in the Methods section on the simulation of blood flow and Lagrangian transport.

- [5] “*How does the non-fluid behaviour of haematocrit in capillaries influence the results?* ”

Blood is a non-Newtonian fluid in that its viscosity depends on tube diameter and haematocrit. This property is described in our simulations using conventional in vivo empirical laws (e.g. Cruz-Hernandez et al. 2019). To test the effect of this property on our results, we have performed additional simulations, where blood was replaced by a Newtonian fluid (Fig. R4 below). The statistics of flow rate, capillary transit time and network travel times are only

weakly affected by this change, which confirms that the non-Newtonian properties of blood do not affect significantly the transport dynamics that we have uncovered. As discussed above for diameter variations (Fig. R3), the robustness of the derived scaling laws results from the fact that flow rate and transit time fluctuations induced by viscosity variations are negligible compared to heterogeneity generated by the random network structure.

Figure R4: **Comparison of flow and transport statistics for blood and a Newtonian fluid**

a) Flow rate PDF. **b)** Vessel transit time PDF. **c)** Network travel time PDF. **a-c)** Scaling laws are computed for a micro-vascular network with (dots) and without RBCs (circles). Quantities with tilde sat on top are rescaled by their average value. The blue dashed line in panel c shows the \mathcal{T}^{-3} for long network travel times.

Change made: We have commented at the end of Section H of the revised supplementary information that the complex rheology of blood does not affect the flow and transport statistics.

- [6] “Sorry for my ignorance, but I find the term local transit time not clearly defined. Is this the transit time of a single vessel? This largely varies with vessel length and the prognostic/diagnostic value of such a definition is not completely clear to me. I understand the relevance of travel times where it would be most efficient if the overall travel times would be the same for all possible pathways. How does this link to the relevance of local transit time?”

Yes, the local transit time is the transit time of a single vessel, defined in Eq. 2 in our manuscript, i.e. as the ratio of the flow velocity by the vessel length. As explained in our answer to item [2], our results show that the large variability of transit times (over about six decades, see inset of Fig. 1 in our manuscript) is mainly due to the heterogeneity of blood flow, with a much smaller influence of vessel diameter and length. Thus, it is not because of

the vessel length variability that such transit times have low prognostic/diagnostic value, but rather because assessing their distribution in the clinics is very difficult by current imaging techniques.

Ultimately, the key property of interest for predictions is the statistics of network travel times, which control the distribution of solute concentrations. Travel times are the sum of successive capillary transit times along blood trajectories. Hence, understanding the distribution of capillary transit times is a key step to predict the distribution of travel times at the scale of the network. In this study, we establish the first mechanistic model (Eq. 6 and 7-9, respectively, in our manuscript) that captures the statistics of capillary transit times and relates it to the statistics of network travel times. As outlined in the discussion, this has been an open question for more than 20 years, so that current models of travel time distributions are purely empirical and, as a result, strongly underestimate the probability of large travel times. The latter control the appearance of critical regions (hypoxic regions and regions with abnormally high concentrations of amyloid- β).

Change made: We have slightly modified the text of footnote 1 of the manuscript to clarify the definition of local transit times by referring to Eq. 2. We also have outlined the fact that travel times can be derived from successions of transit times along each trajectory in Section "Methods", Subsection "Simulation of blood flow and Lagrangian transport (see p. 14 of the revised version).

[7] *Minor:*

Eq.4: Do you have an explanation for $L^=5$? How did you find this number? Which other parameters does it depend on? "*

The scale L^* characterizes the exponential decay of the PDF of trajectory lengths at large lengths $p(L) \sim e^{-L/L^*}$. This exponential decay ends the power law behavior $p(L) \sim L^{-2}$ which is the signature of the dipole topology of intermediate trajectory lengths. It has been estimated by fitting the exponential decay to the PDF obtained from simulations (Fig. 2b). Our analyses with model networks of increasing complexity suggests that the value of L^* is likely controlled by a complex interplay between the topology of the network and the density of venules/arterioles, which prevents arbitrary large trajectories to develop. Hence, we expect this parameter to depend on the topology of the capillary bed, e.g. level of disorder, and/or on the density and structure of venules and arterioles. Establishing precisely this relationship

is the subject of current research.

Change made: We have included this explanation after equation 4 in the revised manuscript.

- [8] “While all other videos played fine, supplementary movie S4 did not.”

Thanks for pointing this out. We have corrected the video.

REPLY TO REVIEWER 3

- [1] “This microvascular physiology-modelling study by the authors Goiran, Le Borgne and Lorthois, from a biological perspective, is an thorough and excellent piece of work on the origins of the microvascular flow heterogeneity in mouse capillary network. In contrast to the classical simplified microvascular simulations used to evaluate the capillary flow patterns and their impact on tissue oxygenation, this model is based on a real and 3-D capillary network and incorporates many physical features of complicated biphasic blood flow dynamics. This study has a great potential to be expanded in the near future for understanding microvascular dysfunction in many acute and chronic neurological diseases and can also be applicable to other tissues outside the nervous system. These results give insight about the origins and impact of the dynamic capillary flow irregularities, like temporary stalls observed in normal animals, that have been shown to be exacerbated in animal models of stroke and Alzheimer’s disease. The extent of the capillary transit variation is highly interesting and also provides clues about the selective vulnerability of deep cortical tissues to microvascular insults. I believe, in overall, this paper is a great contribution to the field.”

We sincerely thank the Reviewer for his/her positive assessment of our work.

- [2] “I just have a few simple comments to improve the discussion of findings.

How applicable are these results to the human microvessel network? A brief discussion on the similarities and differences between mouse and human capillary network architecture can be added to the text to improve the translational value of this work. ”

We thank the reviewer for this suggestion. In fact, mouse and human capillary networks are highly similar (Smith et al. *Frontiers in Physiology*, 2019, ref. 39 in our MS). They are topologically equivalent even if human capillary vessels are 20% longer in average. Moreover, the arteriolar and venular components of the network mainly differ by the average number of arterioles drained by a given venule (Hartmann et al. 2018). Because the observed flow

statistics are generic (see SI section C) and robust (see SI section H), we believe that these differences are not sufficient to alter the underlying physics uncovered in our manuscript, although the parameter values and pre-factors of the associated scaling laws may be slightly dependent on the specific topology of the network.

Reference : Hartmann et al. Does pathology of small venules contribute to cerebral microinfarcts and dementia? *Journal of Neurochemistry*, 2018, 144 (517–526).

Change made: We have enriched the Discussion Section to better highlight that inter-species variability is unlikely to significantly change our conclusions (see end of the discussion on p. 13 of the revised manuscript, where we have included a novel reference to Hartmann, i.e. ref 66).

- [3] “When modeling the oxygen concentration distribution, was different levels of metabolic activity (hence oxygen consumption) across different cortical layers considered? If not, in what way could this modulate the results? ”

Our simplified model for oxygen reactive transport does not account for the different levels of metabolic activity in different cortical layers. Indeed, the parameter k_{O_2} , which represents the first order kinetic constant for oxygen consumption in the brain tissue, is taken as a constant (see third subsection in the Methods Section and Supplementary Information section I). An important point here is that its value is obtained by inverting Eq. 13 to match the typical value of measured arterio-venous oxygen ratios. Thus, imposing a metabolic profile, with the highest metabolic activity in the central layer of the cortex with the highest density of neurons and a decreased metabolic activity in the superficial and deep layers, would not change the average value (i.e. averaged over all trajectories) of oxygen concentration at the outlets of the network. However, it would result in a smaller rate of oxygen decrease along the shortest trajectories ($L = 20 - 30$, which only explore the superficial layers, as well as in the first part ($n/L < 10 - 20\%$) of all longer trajectories, i.e. before the blood trajectories visiting the central layer. Thus, for these trajectories, we expect a higher rate of oxygen decrease until reaching the minimum value, thus exacerbating the “dip” in oxygen concentration observed in the middle of the longest trajectories (Fig. 5c). This will also enhance the differences in venular concentrations observed for trajectories of different lengths (Fig. 5c).

Change made: We have clarified in the Supplementary Information, below Eq. S27, that the

first order kinetic rate constant for oxygen consumption is assumed to be constant across the depth of the cortex.

- [4] *“A strong emphasis is made on Alzheimer pathogenesis, as it is understandable that increased capillary stalling was observed in a mouse AD model and amyloid-beta toxicity is implicated in AD. However, in the paper by Cruz-Hernandez et al, Nat Neuro 2019, PMID 30742116) there was no association between capillaries with abnormal/stalled flow and amyloid deposition. Here, the capillary transit and amyloid beta clearance simulations seem to indicate capillary flow-dynamic hot spots for amyloid species accumulation. What are the authors’ comments on this difference? Moreover, microvascular architecture seems to be changing over the course of AD (as reviewed in Steinman et al, PMID: 33536876). I believe these should be added to the discussion and cited, also discussed how these morphological changes could affect the predictions of capillary flow trajectories and oxygen distribution. ”*

Indeed, no associations have been found between capillaries with abnormal/stalled flow and amyloid deposition in the paper by Cruz-Hernandez et al., Nat Neuro 2019. This is consistent with our modeling framework that shows that the critical regions with abnormally high amyloid- β concentration, which this referee is right to interpret as “*hot-spots*” for amyloid species accumulation, are deterministically located in regions fed by long trajectories. This implies that they are independent on the spatial locations of the capillary stalls. Therefore, our findings provide a new hypothesis to explain the lack of correlation between capillaries with abnormal/stalled flow and amyloid deposition .

Change made: We have added a comment on the deterministic location of hot spots of amyloid species accumulation in the Discussion (p. 12 of the revised version). We have also included a comment at the end of the Discussion (p. 13) about the architectural variability of microvascular networks, which points out to the impact of disease, where we cited the reference suggested by this reviewer (ref. 67).

- [5] *“Should be added to the methods: Which programming platform was used for modeling and mathematical calculations ? And are the models open to the scientific community upon request?”*

The Programming platforms used for modeling and mathematical calculations are the following: flow simulation are performed using an in-house C++ code for High Performance Computing Petsc library (Peyrounette et al. PLOS ONE 2018, ref. 74 in

the revised Manuscript), particle tracking and data analysis was performed using an in-house Python code (V2.7), Laplace inversion of analytical model was performed using Matlab (<https://fr.mathworks.com/matlabcentral/fileexchange/32824-numerical-inversion-of-laplace-transforms-in-matlab>). Numerical datasets as well as post-processing tools will be made available upon request.

Change made: We have added this information in the methods Section (end of first column on p. 14) and in Section A of the Supplementary Information.

REVIEWERS' COMMENTS

Reviewer #1 (Remarks to the Author):

I thank the authors for responding to my comments so clearly and helpfully.

Reviewer #2 (Remarks to the Author):

Thank you very much for detailed explanations and clarification. I am happy with the modifications.

Reviewer #3 (Remarks to the Author):

All my comments have been properly addressed. I thank the authors for their contribution to the field.

We thank the reviewers for their very positive feedbacks. There was no remaining concerns from the reviewers, who all were satisfied with our responses and revisions.

Reviewer #1 (Remarks to the Author): I thank the authors for responding to my comments so clearly and helpfully.

Reviewer #2 (Remarks to the Author): Thank you very much for detailed explanations and clarification. I am happy with the modifications.

Reviewer #3 (Remarks to the Author): All my comments have been properly addressed. I thank the authors for their contribution to the field.